# High Doses of Inhaled Nitric Oxide as an Innovative Antimicrobial Strategy for Lung Infections

**DOI:** 10.3390/biomedicines10071525

**Published:** 2022-06-28

**Authors:** Lorenzo Del Sorbo, Vinicius S. Michaelsen, Aadil Ali, Aizhou Wang, Rafaela V. P. Ribeiro, Marcelo Cypel

**Affiliations:** Latner Thoracic Research Laboratories, Toronto General Hospital Research Institute, University Health Network, University of Toronto, 101 College Street | PMCRT, Toronto, ON M5G 1L7, Canada; lorenzo.delsorbo@uhn.ca (L.D.S.); aadil.ali@uhn.ca (A.A.); aizhou.wang@uhn.ca (A.W.); rafaela.vaninpintoribeiro@uhn.ca (R.V.P.R.); marcelo.cypel@uhn.ca (M.C.)

**Keywords:** nitric oxide, inhaled nitric oxide administration, antibacterial effect, antiviral effect, lung infections

## Abstract

Since the designation of nitric oxide as “Molecule of the Year” in 1992, the scientific and clinical discoveries concerning this biomolecule have been greatly expanding. Currently, therapies enhancing the release of endogenous nitric oxide or the direct delivery of the exogenous compound are recognized as valuable pharmacological treatments in several disorders. In particular, the administration of inhaled nitric oxide is routinely used to treat patients with pulmonary hypertension or refractory hypoxemia. More recently, inhaled nitric oxide has been studied as a promising antimicrobial treatment strategy against a range of pathogens, including resistant bacterial and fungal infections of the respiratory system. Pre-clinical and clinical findings have demonstrated that, at doses greater than 160 ppm, nitric oxide has antimicrobial properties and can be used to kill a broad range of infectious microorganisms. This review focused on the mechanism of action and current evidence from in vitro studies, animal models and human clinical trials of inhaled high-dose nitric oxide as an innovative antimicrobial therapy for lung infections.

## 1. Introduction

Nitric oxide (NO) is an endogenous free-radical gas produced by numerous cell types in the body with several important direct or indirect biological roles in processes such as cellular signalling in neurotransmission, vasodilation, inhibition of platelet activation, as well as leukocyte adhesion and host defense [1,2,3,4,5].

Due to these properties, therapies that enhance the release of endogenous NO or directly deliver exogenous NO have been developed to treat several diseases [6,7,8]. In particular, given NO’s ability to rapidly diffuse through biological fluids and lipid membranes, its delivery through inhalation has been studied in the treatment of respiratory diseases. In clinical practice, inhaled nitric oxide (iNO) is currently used as a selective pulmonary vasodilator to reduce right ventricular afterload in patients with severe right ventricular failure. It is also used to reduce pulmonary ventilation–perfusion mismatch in patients with refractory hypoxemia due to acute respiratory distress syndrome or after lung transplantation [9,10,11,12]. This selective vasodilatory effect on the pulmonary circulation is achieved by delivering doses of iNO in the range of 10–80 parts per million (ppm).

When delivered at doses greater than 160 ppm, iNO has been described as having antimicrobial properties and can be potentially used to kill a broad range of infectious organisms [13,14,15,16], including severe acute respiratory syndrome coronavirus 2 (SARS-CoV-2) [17]. These findings resulted in an increased number of studies during the pandemic aimed at exploring the therapeutic antimicrobial use of iNO [18,19,20,21,22,23,24,25,26,27,28,29,30,31,32,33,34,35].

This narrative review focuses on the mechanism of action and current uses of high-dose inhaled nitric oxide as antimicrobial therapy for lung infections.

## 2. Biosynthesis of Nitric Oxide

NO is a free radical and cellular signalling molecule synthesized from the nitrogen group of the amino acid L-arginine by an enzyme, nitric oxide synthase (NOS). Several isoforms of this enzyme have been characterized [36]. Although the isoforms catalyze the same reaction, they differ with respect to their regulation, amplitude and duration of NO production, and cellular and tissue distribution [14]. These isoforms are usually classified as either constitutive or inducible. The constitutive NO synthase (cNOS) is calcium-dependent and releases NO in low concentrations for brief periods in response to a receptor or physiological stimulation. The cNOS acts as a transduction-signalling molecule, underlying several physiological responses. The constitutive forms include endothelial NO synthase (eNOS/NOS3), which mediates endothelium-dependent vasodilator responses, and neuronal NO synthase (nNOS/NOS1). The other isoform is inducible NO synthase (iNOS/NOS2), which is activated in response to cytokines and specific substances related to infection like endotoxins. Once activated and expressed, iNOS synthesizes NO in large amounts for a prolonged time. This isoform is calcium-independent but requires other cofactors to work [37]. iNOS has been shown to contribute to nonspecific defense mechanisms against invading microorganisms and tumor cells. These NOS isoforms have also been identified in the respiratory system as sources of endogenous NO in the airways, such as in the pulmonary endothelial cells, inflammatory cells, fibroblasts, smooth muscle cells and nitrergic nerves [38,39,40,41].

## 3. Nitric Oxide Concentration-Dependent Mechanisms of Action

### 3.1. Low-Dose NO as a Molecular Mediator of Cellular Signalling

The activity and mechanism of action of NO are concentration-dependent. The endogenous production of NO falls into two categories depending on the expression of the different isoforms of the NOS enzyme. Synthetized at low doses (basal levels) by the cNOS isoform, NO works as a cellular-signalling biomolecule and functions predominantly in the regulation of vascular tone [5]. Most of the NO-induced vascular effects are mediated by the release of cyclic guanosine monophosphate (cGMP). Cyclic guanosine monophosphate results from the activation of the soluble guanylate cyclase by NO, which then transforms guanosine triphosphatase into cGMP [42]. Based on these mechanisms, Roberts et al. hypothesized that the delivery of inhaled NO would allow for its diffusion through the lung and selectively vasodilate the pulmonary vessels [43]. Their experiments supported the hypothesis, resulting in a seminal publication in respiratory medicine. Warren Zapol, Jesse D. Roberts and their team were indeed the pioneers that overturned the perception of NO from a highly toxic gas into a therapeutic molecule, now approved for clinical use by the FDA. They were the first to propose iNO as a clinical therapy for near-term babies with pulmonary hypertension. Over 150,000 babies in the US alone have benefited from iNO, and more than 350,000 adults and children have received it for off-label uses, including the treatment of pulmonary hypertension, acute respiratory distress syndrome and many other diseases [11].

### 3.2. High-Dose NO and Its Antimicrobial Effect

When synthesized by the inducible isoform of NOS (iNOS), NO is generated at very high concentrations, 10–100 times higher than the cNOS. Once activated, iNOS generates sustained high levels of NO over many hours. In this condition, high concentrations of NO can promote microbicidal biochemical reactions, participating in the host defense mechanisms of the innate immune response [44,45]. Two of the essential antimicrobial systems of the innate immunity are the iNOS and the nicotinamide adenine dinucleotide phosphate (NADPH) oxidase pathways. While the product of the iNOS activity is NO, NADPH oxidase generates superoxide (O_2_^−^). Spontaneous or catalyzed reactions involving NO and O_2_^−^ result in the generation of highly reactive intermediate molecules, such as H_2_O_2_, OH^−^, singlet oxygen, hypochlorous acid, nitrogen dioxide (NO_2_), peroxynitrite (ONOO^−^), dinitrogen trioxide, dinitrosyl iron complexes, nitrosothiols or nitroxyl, which are named reactive nitrogen species (RNS) and reactive oxygen species (ROS). RNS and ROS are characterized by different stability, have a complex biological role, can be simultaneously present and are both key effectors of the antimicrobial activity of the innate immunity. In particular, ROS and RNS can interact with numerous targets in microbial cells, including thiols, metal centres, protein tyrosines, nucleotide bases and lipids, leading to microbial cell damage [46,47]. Therefore, ROS and RNS might ultimately halt and suppress pathogen replication through mechanisms that involve nitrosation, nitrosylation and nitrosative and oxidative damage of microbial DNA, RNA, proteins, lipids and membranes. They can also interfere with replication, respiratory chain, inhibition of enzymes of synthesis and DNA repair. These mechanisms are outlined in Figure 1. 

Although NO is likely to attack multiple structures of the infectious pathogen simultaneously, the disruption of the iron–sulphur bond (Fe-S) alone can be sufficient for a strong antimicrobial effect. The antimicrobial actions of RNS have been demonstrated in a variety of organisms, such as fungi, parasites and bacteria [16]. In contrast, conflicting results have been reported concerning the role of NO against viral infections [48].

## 4. High-Dose Exogenous NO as an Antimicrobial Treatment

The description of the antimicrobial effects of the endogenous NO at high concentrations led to the development of studies investigating the potential antimicrobial effects of exogenous NO at high doses [49,50,51].

### 4.1. High-Dose Exogenous Nitric Oxide: In Vitro Studies

Evidence from in vitro studies contributed to the definition and characterization of the antimicrobial effect of NO. Importantly, it was possible to determine in vitro the appropriate concentration of NO with antimicrobial effects, which has been subsequently adopted in clinical studies. Ghaffari et al. identified the lowest effective antimicrobial dose of NO, showing that in vitro continuous exposure to NO at doses greater than 160 ppm for up to 6 h was able to eliminate bacterial colonies (10^5^ CFU/mL) of *Escherichia coli, Pseudomonas aeruginosa* and *Staphylococcus aureus.* Doses of NO below 160 ppm induced only a bacteriostatic effect, which seemed to be dose-dependent [13,52]. These findings were confirmed in studies using high-dose NO (200 ppm) on multi-drug-resistant strains of microorganisms, including methicillin-resistant *Staphylococcus aureus* and resistant *Pseudomonas aeruginosa* [52,53,54,55].

Due to concerns regarding the potential in vivo toxicity of continuous exposure to high-dose NO, in vitro investigations were conducted to identify alternative efficacious regimens. Interestingly, the bactericidal effect was retained with intermittent delivery of high-dose NO for 30 min every 3.5 h for a total of 15 h. Therefore, an extension of the total treatment duration was required to achieve the same antimicrobial effect obtained with continuous delivery (Figure 2) [56].

The antimicrobial effect of exogenous NO has also been studied in vitro against viruses. NO donors, compounds able to release NO at the desired concentration, or substrates of NO synthesis, such as L-arginine, have been shown to inhibit various types of viruses at different stages of replication [57,58]. Evidence of NO-induced inhibition of viral replication has been reported for some DNA viruses, such as hepatitis B and herpes simplex, and for some RNA viruses, such as influenza A, influenza B and coronavirus [48,59,60,61,62]. Croen et al. performed one of the first studies reporting the antiviral effect of NO using NO donors. S-nitroso-L-acetylpenicillamine (SNAP), an exogenous NO donor, added in vitro (500 μM) to the media of a monolayer of cells 3 h after the infection with herpes simplex virus type 1 (HSV 1), induced a 105 ± 54-fold reduction of viral replication, which was significantly higher compared with control media. Additional experiments showed that the NO-induced HSV 1 replication inhibition in three different types of infected cells (Vero, Hep2 and RAW 264.7) was dose-dependent [63]. Similar results, demonstrating significant NO-induced inhibition of viral replication, were obtained with SNAP in another in vitro study in Mabin Darby canine kidney (MDCK) cells infected with either influenza A (A/Netherlands/202/95 [H3N2]) or influenza B (B/Netherlands/22/95) viruses. The strongest inhibition was observed by using SNAP at a concentration of 400 μM [64]. Further experiments also demonstrated the antiviral efficacy of NO delivered in gaseous form. MDCK cells, infected with one of three different strains of influenza (influenza A viruses, A/Denver/1/19 57 (H1N1), A/Victoria/3/75(H3N2) and influenza B Virus, B/Hong Kong/5/72), were exposed to NO gas mixture of 160 ppm, delivered continuously for up to 2.5 h and compared to control cells, which were infected but exposed only to air. The results showed that, in this model, NO was able to significantly inhibit the infectivity of influenza A viruses but not influenza B. However, when virions were exposed to NO at concentrations of 80 and 160 ppm for 10–120 min prior to infection, complete inhibition of infectivity was achieved for all three viral strains (Figure 3) [65].

Importantly, recent experimental evidence proved the in vitro antiviral effect of NO also against the SARS-CoV-2 [17]. SNAP, administered at doses of 200 μM or 400 μM as NO donor, induced in Vero E6 cells infected with SARS-CoV-2 a dose-dependent inhibition of viral replication. However, in this model, the viral replication was not completely eliminated even with the highest dose of SNAP. Nonetheless, the treatment with SNAP resulted in the reduction or elimination of the viral-induced cytopathic effects.

### 4.2. High-Dose Exogenous Nitric Oxide: In Vivo Experimental Studies

The antibacterial and antiviral effect of high-dose iNO has been investigated in a few animal experimental models. In a rat pneumonia model caused by *Pseudomonas aeruginosa*, the intermittent administrations of iNO at 160 ppm for 30 min every 4 h for a total of 12 or 24 h was well tolerated with no relevant side effects and significantly reduced bacterial colony counts in the lungs compared with controls [66]. In another study, to investigate the possible dose-response effect of iNO, mice underwent lung infection with *Klebsiella pneumoniae*, and were then treated with either iNO at 80, 160 or 200 ppm continuously for 48 h or air as control. An additional experimental group received 300 ppm iNO delivered intermittently for 12 min every 3 h for the entire 48 h duration of the experiment. Compared to controls, a significant reduction in mortality and bacterial counts in lung and spleen was observed only in animals receiving intermittent iNO at 300 ppm. These results suggest that the right balance between dose and exposure time may play an important role in the antimicrobial effectiveness of iNO [67]. The intermittent and continuous regimens of iNO administration have also been tested in an in vivo experimental model of influenza A infection in C57B1/6 mice. In this model, however, the prophylactic (1 h prior infection) or therapeutic (4 h post-infection) iNO administration at 80 ppm in a continuous fashion or 160 ppm for 30 min every 3.5 h up to 12 days failed to decrease the mortality rate. In addition, both regimens failed to decrease the viral load of mice infected with influenza in the lungs [68].

### 4.3. High-Dose Exogenous Nitric Oxide: Clinical Studies

The clinical administration of high-dose NO as a potential antimicrobial therapy has been increasingly studied. These studies focused on the effect of NO on the lung due to the ease of its selective delivery through the inhaled route.

The pulmonary delivery of high-dose iNO can potentially cause unwanted toxic effects due to its highly reactive nature. NO can in fact rapidly bind hemoglobin with high affinity to form methemoglobin (metHb). Persistent high levels of metHb (>6%) are associated with poor delivery of oxygen (O_2_) to tissues. In addition to metHb formation, high-dose iNO, when co-delivered with O_2_, can generate toxic compounds, such as nitrogen dioxide (NO_2_). To circumvent this problem, as previously mentioned, researchers have explored iNO delivery regimens using intermittent high doses for short periods of time (for example, 160 ppm for 30 min) followed by a recovery window in between treatments (minimal 3.5 h) to avoid significant methemoglobinemia and reduce the potential toxicity. Miller et al. performed the first human clinical study to evaluate the feasibility and safety of the intermittent protocol [69]. They delivered to healthy adult volunteers 160 ppm of iNO for 30 min, five times per day, for 5 consecutive days. There was no evidence of lung injury or airway inflammation based on lung function parameters and key pro-inflammatory cytokines measured in plasma. Likewise, no significant adverse events were reported, and all individuals tolerated the iNO treatment, demonstrating the feasibility and safety of this high-dose iNO delivery regimen. Deppish et al. used the same intermittent high-dose iNO delivery strategy to treat antibiotic-resistant bacterial and fungal lung infections in patients with cystic fibrosis (CF). This study confirmed the safety of this treatment and demonstrated that high-dose iNO considerably reduced the lung microbial load and inflammation in these patients, resulting in improved lung function. However, in Deppish’s study, the intermittent high-dose iNO delivery regimen failed to achieve complete microbial eradication, suggesting a large potential for improvement in defining the optimal dose and treatment duration of in vivo antimicrobial strategies with high-dose iNO [70]. A similar conclusion was reached by Bentur et al. in a recent pilot study in CF patients with refractory *Mycobacterium abscessus* (non-tuberculous mycobacteria—NTM) lung infection. In this study, the intermittent delivery of 160 ppm iNO for a total of 21 days induced only a reduction and not the microorganism’s complete eradication of NTM in the lung. The investigators hypothesized that failure to achieve eradication could be attributed to potential NO neutralization, suggesting that an even longer duration of iNO treatment may be necessary to achieve eradicating bactericidal activity. The authors also hypothesized that prolonged intermittent iNO treatment beyond 3 weeks could increase the susceptibility of NTM biofilm to antibiotics and achieve improved bactericidal activity in CF airways [71]. However, more recent case reports showed that even longer durations (up to 29 days) and higher doses of iNO (up to 250 ppm) failed to induce the eradication of infective microorganisms [72,73]. 

Overall, the in vivo delivery of exogenous high-dose iNO reported in the above-described clinical trials did not show the same remarkable antibacterial effect observed in the in vitro studies, despite the high variability of applied doses and treatment duration of the intermittent administration strategies. These data suggest that new research should aim at proving in vivo safety and efficacy of the high-dose iNO regimens effective in vitro, with continuous rather than intermittent delivery and prolonged rather than short exposure time.

In Table 1, we summarize the most relevant in vitro, in vivo and clinical studies regarding antibacterial high-dose NO treatment.

Several studies also investigated the antiviral effect of iNO with promising results. In particular, a multicenter randomized pilot clinical trial tested the safety and tolerability of intermittent high-dose iNO delivery in hospitalized infants with viral bronchiolitis. Each patient received intermittent high-dose iNO (160 ppm) plus oxygen/air for 30 min or oxygen/air alone (control), five times/day, up to 5 days. Sixty-nine infants were enrolled in this study. The results showed that the treatment with intermittent iNO was well tolerated, and the secondary clinical endpoints suggested a benefit [76]. However, the lack of a larger clinical trial with patient-centered outcome as the primary endpoint precludes any further conclusion concerning iNO treatment in this patient population. More recently, due to the dramatic impact of the SARS-CoV-2 pandemic on the health system, several case series reported data on the treatment with iNO in patients with coronavirus disease 2019 (COVID-19) [77]. In all these studies, high-dose iNO (160–200 ppm) was delivered with intermittent regimens for 30 min per session twice daily. Overall, it was reported that the treatment was well tolerated and that it resulted in improved oxygenation and cardiopulmonary function, without evidence of toxicity or adverse events [18,20,27].

However, due to the observational nature of these case reports and the overall small number of COVID-19 patients treated with high-dose iNO, these data are hypothesis-generating, but insufficient to suggest any clinical benefit. Important questions regarding the definition of the regimen with the highest efficacy and the lowest toxicity remain open.

## 5. Novel In Vivo Strategies to Deliver Continuously Antimicrobial Doses of Inhaled Nitric Oxide

In an attempt to optimize treatment efficacy, our group performed a number of experimental studies to investigate the hypothesis that a regimen of prolonged (6–12 h) continuous instead of intermittent iNO delivery could have a superior antimicrobial effect. These studies investigated the safety profile of this regimen due to the potential increased iNO-induced toxicities, including a higher risk of methemoglobin and NO_2_ formation, hemodynamic changes and platelet function alteration [78].

### 5.1. Continuous Inhaled Nitric Oxide Delivery during Ex-Vivo Lung Perfusion

In order to investigate whether prolonged (12 h) continuous delivery of high-dose iNO (200 ppm) is feasible and not injurious to the lung parenchyma, this regimen was applied to an ex vivo model of perfused and ventilated lung. 

Ex vivo lung perfusion (EVLP) is a novel platform developed by our group, which allows for the delivery of therapeutic agents to the lung in an isolated environment (Figure 4), therefore eliminating the risk of collateral side effects to other peripheral organs [79,80]. It has been used clinically in our center over the past decade to allow a more objective assessment of donor lung function, thereby successfully increasing the use of donor lungs for transplantation [81]. 

EVLP has also been used to study innovative therapies for lung regeneration and modification, such as surfactant replacement, ABO antigen depletion, α1 antitrypsin administration, gene therapies, photodynamic therapies, ultraviolet C irradiation and high-dose antibiotics [82,83]. 

In the Toronto EVLP platform, lungs are perfused with an acellular solution. Therefore, the absence of blood in the system provides the opportunity to investigate the pure isolated effects of NO and NO_2_ on the parenchyma of healthy lung and evaluate any potential lung toxicity related to high-dose inhaled nitric oxide, unrelated to its potential effect on the formation of metHb.

In a porcine EVLP model, we demonstrated that the continuous administration of high-dose iNO at 200 ppm for 12 consecutive hours was feasible and did not cause lung injury [84]. In our study, healthy lungs were procured from male Yorkshire pigs (28–35 kg), flushed with the standard lung preservation solution and placed on the EVLP platform using the Toronto EVLP technique [85]. The lungs were randomized into two groups (*n* = 4/group): (1) control and (2) treatment with continuous iNO at 200 ppm. Physiologic and biologic measures of lung injury were monitored for 12 h. We found that important physiologic parameters such as oxygenation, compliance and airway pressure remained stable and were comparable to those of controls. In addition, evaluation of lung histology showed no histologic evidence of injury in the treated lungs. Furthermore, no significant inflammatory response was induced by iNO treatment in comparison to controls, as indicated by there being no significant increases in pro-inflammatory cytokines, as outlined in Figure 5.

This was the first time that high-dose NO was continuously delivered for an extended period of time without presenting signs of toxicity, inflammation or pulmonary deterioration [84].

### 5.2. Continuous Inhaled Nitric Oxide Delivery in an In Vivo Large Animal Model

We further investigated the safety of the continuous delivery of high-dose iNO for a protracted period of time in an in vivo large animal model to specifically address the potential systemic toxicity induced by iNO and the formation of metHb [86]. The experimental protocol included the possibility of administrating methylene blue (MB) intravenous (IV) as a reducing agent to treat methemoglobinemia and reconvert the ferric heme of methemoglobin (Hb (Fe (III))) into the functional ferrous state Hb (Fe(II)) [87,88,89].

Yorkshire pigs were sedated, mechanically ventilated and randomized to one of the following two groups: (1) control and (2) treatment with continuous iNO 160 ppm + methylene blue (MB), 1 mg/Kg, as an IV bolus, whenever it was required to maintain metHb <6%. Both groups were ventilated continuously for 6 h; then, the animals were weaned from sedation and mechanical ventilation and followed for 3 days. During treatment and on the third post-operative day, serial assessments were performed to assess lung function and potential pulmonary or systemic injury. No significant change in lung function or inflammatory markers was observed during the study period. Gas exchange, histology and lung tissue cytokines were similar in treated and control animals. 

During the administration of iNO, increased levels of metHb (maximum level 6.9%) were observed in every treated animal, which was then rapidly decreased by MB treatment. One dose of MB was sufficient to maintain levels of metHb <6%. Levels of NO_2_ remained below the safety threshold (5 ppm) during the entire experiment, with an average level of 3.24 ± 0.35 ppm. No other significant changes were observed in blood biochemical markers. Our findings showed that continuous 6 h delivery in vivo of 160 ppm iNO with adjuvant MB is feasible and safe (Figure 6) [86]. 

Although our in vivo study did not focus on the antimicrobial effect of iNO, it generated pre-clinical safety data to design a pilot clinical trial (NCT 04383002) investigating the efficacy of continuous delivery of high-dose iNO as antimicrobial therapy, including treatment for patients with COVID-19. This was particularly timely, as several centers in the last two years have launched clinical trials (NCT 04306393, 03331445, 04338828, 04305457, 04476992, 04312243) investigating the feasibility and efficacy of high-dose iNO for the treatment of COVID-19 [20,90]. However, except for the trial from our center, the protocols of all the other studies were designed with intermittent rather than continuous delivery of iNO.

## 6. Conclusions and Future Perspectives

The potential of iNO as an antimicrobial therapy has gained significant interest in the scientific community and the clinical field. As summarized in this review, increasing evidence suggests that the delivery of high-dose inhaled NO for its significant antimicrobial activity may be safe and could have promising clinical therapeutic applications in the treatment of lung infections. 

The development of this innovative antimicrobial therapeutic strategy is important for several reasons. First, this strategy utilizes novel mechanisms of action to achieve antimicrobial activity, different from those of the current conventional antibiotic medications. Second, the iNO antimicrobial effect is independent of the microbial resistance to the conventional therapeutic agents, whose extensive use is causing the progressive global development of potentially lethal multi-drug-resistant species. Third, the treatment with high-dose iNO is potentially effective against multiple pathogen species and polymicrobial infections. Furthermore, high-dose iNO has a potential direct antiviral effect against respiratory viruses, hence representing a therapeutic option for viral lung infections, including COVID-19. 

Future studies are needed to further elucidate the optimal regimen of high-dose iNO to maximize the antimicrobial effect while minimizing its potential toxicity. In particular, essential issues concerning continuous vs. intermittent delivery, effective antimicrobial dose range, timing of administration and treatment duration warrant specific investigations. Furthermore, high-quality clinical evidence is still required to conclusively establish the efficacy of high-dose iNO delivery in improving outcomes in patients with infections of the respiratory system.

## Figures and Tables

**Figure 1 biomedicines-10-01525-f001:**
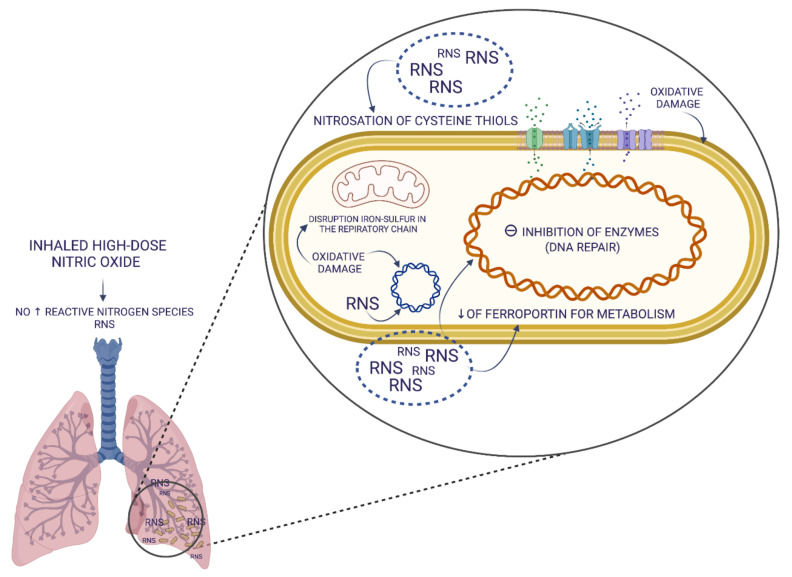
Schematic representation of multiple target structures of the antimicrobial actions of the reactive nitrogen species, which can kill or halt the growth of microorganisms. Definition of abbreviations: NO = nitric oxide, RNS = reactive nitrogen species. The figure was created with BioRender.com (accessed on 19 March 2022).

**Figure 2 biomedicines-10-01525-f002:**
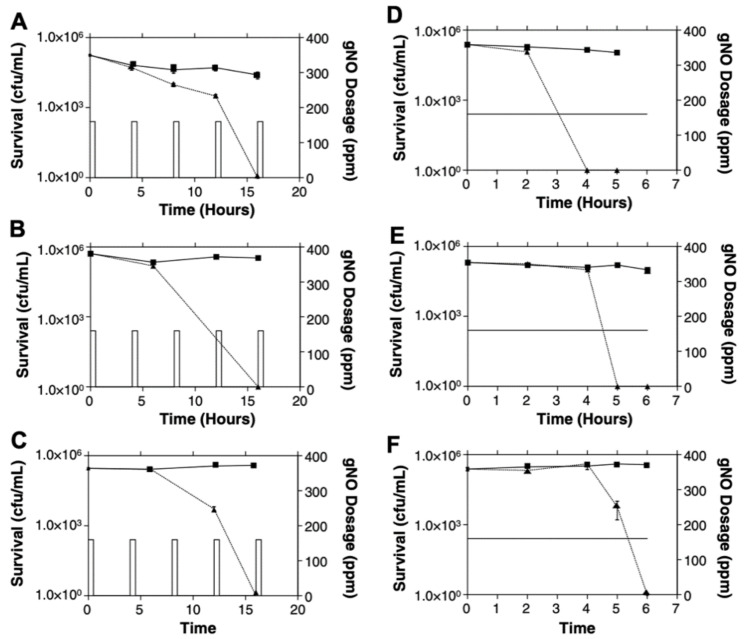
Bacterial survival curves during treatment with intermittent high-dose NO (**A**–**C**) or continuous high-dose NO (**D**–**F**). Bacteria were exposed to intermittent high-dose NO (160 ppm) for 30 min every 3.5 h for a total of 15 h ((**A**–**C**) triangles) or continuous 160 ppm NO ((**D**–**F**) triangles) for 6 h and compared to controls treated with medical air (squares). (**A**,**D**) *Staphylococcus aureus*; (**B**,**E**) *Pseudomonas aeruginosa* (cystic fibrosis clinical strain); (**C**,**F**) *Escherichia coli*. Reproduced with permission [56].

**Figure 3 biomedicines-10-01525-f003:**
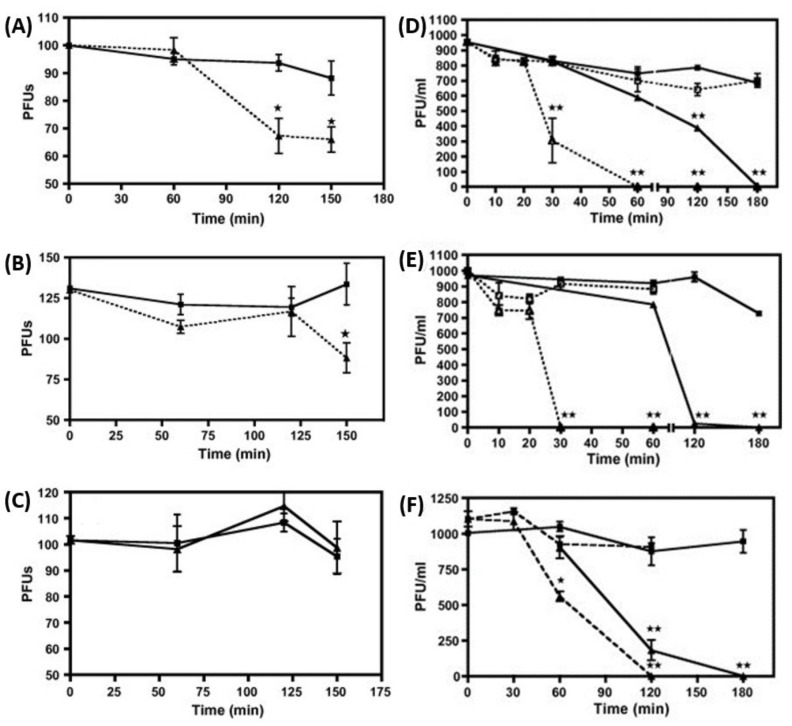
Inhibition effect of NO on influenza virus. (**A**,**D**) H1N1, (**B**,**E**) H3N2, (**C**,**F**) Influenza B. **Left Graphics**: Confluent cells were infected with the indicated virus at 100 PFU/well and then treated with 160 ppm NO for up to 2.5 h (triangles) or medical air as control (squares). Virus infectivity was assessed by plaque reduction assay. **Right Graphics**: Virions (1000 PFUs) suspended in normal saline were treated prior to infection with 80 or 160 ppm NO for 10–120 min (triangles) or medical air as control (squares). At the end of the treatment, the suspensions were used to infect a monolayer of confluent cells. Infectivity was assessed by plaque-reduction assay. * or ** *p* < 0.05. Reproduced with permission [65].

**Figure 4 biomedicines-10-01525-f004:**
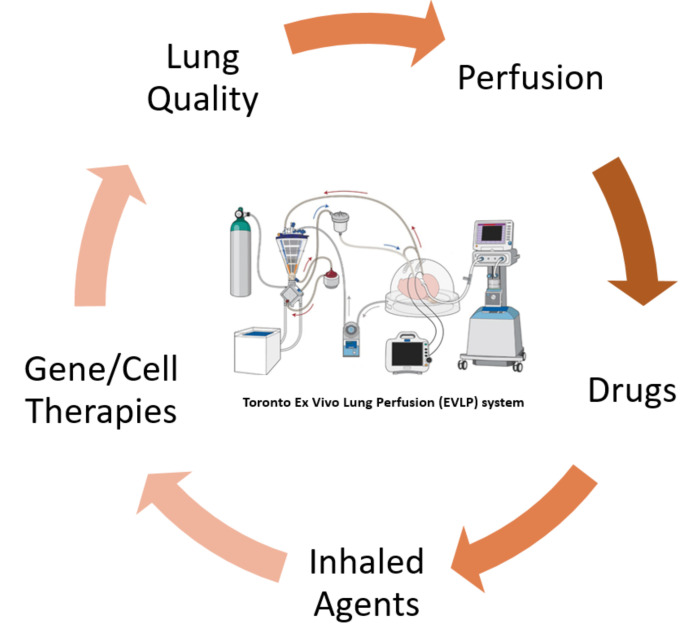
Concept illustration of the normothermic ex vivo lung perfusion (EVLP) platform and opportunities for treatment strategies. Adapted from [79,82].

**Figure 5 biomedicines-10-01525-f005:**
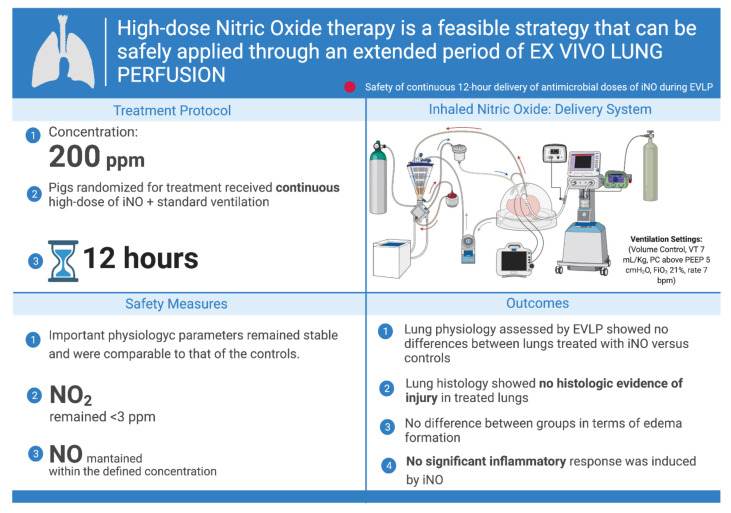
Visual summary of the safety phase study using continuous antimicrobial doses of inhaled nitric oxide during ex vivo lung perfusion. Definition of abbreviations: ppm = parts per million, NO = nitric oxide, iNO = inhaled nitric oxide, NO_2_ = nitrogen dioxide, VT = tidal volume, PEEP= positive end-expiratory pressure, bpm = beats per minute, FiO_2_ = fraction of inspired oxygen. The figure was created with BioRender.com (accessed on 19 March 2022).

**Figure 6 biomedicines-10-01525-f006:**
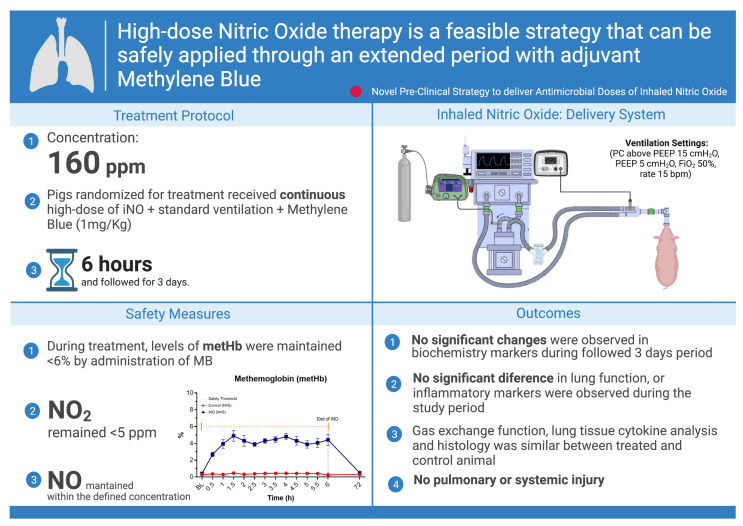
Visual summary of the feasible strategy to safely deliver in vivo high doses of nitric oxide with adjuvant methylene blue. Definition of abbreviations: ppm = parts per million, NO = nitric oxide, iNO = inhaled nitric oxide, metHb = methemoglobin, NO_2_ = nitrogen dioxide, MB = methylene blue, PC = pressure control, PEEP = positive end-expiratory pressure, bpm = beats per minute, FiO_2_ = fraction of inspired oxygen. The figure was created with BioRender.com (accessed on 22 June 2022).

**Table 1 biomedicines-10-01525-t001:** Experimental selected protocols of antibacterial high-dose nitric oxide delivery.

Study	Model	Dose	Treatment Protocol	Year
Ghaffari et al. [13]	in vitro	160–200 ppm	Continuous 48 h	2005
McMullin et al. [55]	in vitro	200 ppm	Continuous 5 h	2005
Miller et al. [56]	in vitro	160–200 ppm	Intermittently 30 min every 3.5 h and Continuous for up to 24 h	2009
Miller et al. [66]	in vivo animal model	160 ppm	Intermittently 30 min every 4 h	2013
Wiegand S. et al. [67]	in vivo animal model	160/200 ppm 300 ppm	Continuous (48 h) Intermittent 12 min, every 3 h for 48 h	2021
Miller et al. [69]	Clinical Study	160 ppm	Intermittently 30 min every 3.5 h, 5 times daily	2012
Deppish et al. [70]	Clinical Study	160 ppm	Intermittently 30 min, 3 times daily of 5 days	2016
Yaacoby-Bianu et al. [74]	Compassionate	160 ppm	Intermittently Minimal time interval 3.5 h (max 21 days)	2018
Bentur et al. [71]	Pilot Clinical Study	160 ppm	Intermittently 30 min, 5 times daily for 14 days and 3 times daily for 7 days	2020
Bartley et al. [75]	Clinical Study	160 ppm	Intermittent over a 28-days	2020
Bogdanovski et al. [72]	Compassionate	up to 240 ppm	Intermittently two courses 5 times daily for 5 days and 3 times daily for 8 days	2020
Goldbart et al. [76]	RCT	160 ppm	Intermittently 30 min, 5 times daily of 5 days	2020
Goldbart et al. [73]	Compassionate	150–250 ppm	Intermittent 4 times a day for 2 weeks, 2 times a day 2 weeks and one in the last day of treatment. (29-day treatment course)	2021

## Data Availability

Data sharing not applicable.

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
