# Peer review of "High Doses of Inhaled Nitric Oxide as an Innovative Antimicrobial Strategy for Lung Infections"

_biomedicines, 2022, doi:10.3390/biomedicines10071525_

Round 1

Reviewer 1 Report

The manuscript biomedicines-1726323 is a review on the role of nitric oxide as an antimicrobial. The topic is original and the information contained in the review is updated and relevant for the research in this field.

I consider that this manuscript is appropriate for publication in Biomedicines after little revision.

I particularly appreciated that the authors cited the generation of reactive oxygen species (ROS) in the mechanisms of action, but I was a bit disappointed that this aspect has not been very developed.

The Author stated that “the in vivo delivery of exogenous high-dose iNO, reported in the above described clinical trials, did not show the same remarkable antibacterial effect observed in the in vitro studies”. This deserves to be discussed, at least a little.

At the end of the manuscript there is a whole part highlighted in yellow that has not been compiled.

Citations in References must be modified according to the style indicated by the journal (see Author Guidelines).

I suggest a rereading by the Authors, to resolve some typos.

Author Response

Toronto, May 21st, 2022

To the Editor of Biomedicines

Biomedicines-1726323: “High doses of inhaled nitric oxide as an innovative antimicrobial strategy for lung infections”

Dear Editor, Dear Reviewer,

Many thanks for your thoughtful comments and review of our work.

We have revised the manuscript according to your concerns.

Please find below our point-by-point response to your review letter. We are most grateful for the possibility to revise the manuscript. We hope that all changes and corrections are in accordance with your concerns, which were very helpful in improving the quality of our work.

Sincerely yours,

Vinicius Schenk Michaelsen

REVIEWER 1

Comments and Suggestions for Authors

The manuscript biomedicines-1726323 is a review on the role of nitric oxide as an antimicrobial. The topic is original and the information contained in the review is updated and relevant for the research in this field.

Thanks for the comment, we really appreciate it.

I consider that this manuscript is appropriate for publication in Biomedicines after little revision.

We are grateful for your kind comment.

I particularly appreciated that the authors cited the generation of reactive oxygen species (ROS) in the mechanisms of action, but I was a bit disappointed that this aspect has not been very developed.

We thank the reviewer for this suggestion. We modify the manuscript (see section 3.2), adding a paragraph to better describe the role of ROS in the mechanisms of action of nitric oxide. However, since ROS are generated by NAPH oxidase and not by iNOS, we decided not to extensively describe the role and mechanisms of ROS in the host defense mechanisms, as not part of the specific scope of our review.

The Author stated that “the in vivo delivery of exogenous high-dose iNO, reported in the above described clinical trials, did not show the same remarkable antibacterial effect observed in the in vitro studies”. This deserves to be discussed, at least a little.

We thank the reviewer for this comment. We further discussed this issue in the revised version of the manuscript (see section 4.3)

At the end of the manuscript, there is a whole part highlighted in yellow that has not been compiled.

Thanks for bringing this issue to our attention. This part of the submitted manuscript has now been modified according to the editorial instructions.

Citations in References must be modified according to the style indicated by the journal (see Author Guidelines).

Thanks for this comment. We changed the reference style according to the editorial indications.

I suggest a rereading by the Authors, to resolve some typos.

We apologize for the typos, which are now corrected.

Reviewer 2 Report

The work entitled “High doses of inhaled nitric oxide as an innovative antimicrobial strategy for lung infections” deals with the mechanisms of action and current perspectives of inhaled high- dose nitric oxide as an innovative antimicrobial therapy for lung infections. The work is extremely well organized and put together. There some English mistakes that must be corrected along the manuscript but nothing too important that takes way from the authors’ merit. The authors should also pay attention to the verbal time: for instance they should not use “will focus” in the abstract but rather “focused” because the review is already done.

Regardless, the work is scientifically sound, the schematics provided by the authors are original and very interesting. There are also somethings missing from the article formatting but the authors may have know that already since they left those sections highlighted in yellow.  

Author Response

Toronto, May 21st, 2022

To the Editor of Biomedicines

Biomedicines-1726323: “High doses of inhaled nitric oxide as an innovative antimicrobial strategy for lung infections”

Dear Editor, Dear Reviewer,

Many thanks for your thoughtful comments and review of our work.

We have revised the manuscript according to your concerns.

Please find below our point-by-point response to your review letter. We are most grateful for the possibility to revise the manuscript. We hope that all changes and corrections are in accordance with your concerns, which were very helpful in improving the quality of our work.

Sincerely yours,

Vinicius Schenk Michaelsen

REVIEWER 2

The work entitled “High doses of inhaled nitric oxide as an innovative antimicrobial strategy for lung infections” deals with the mechanisms of action and current perspectives of inhaled high- dose nitric oxide as an innovative antimicrobial therapy for lung infections. The work is extremely well organized and put together. There some English mistakes that must be corrected along the manuscript but nothing too important that takes way from the authors’ merit. The authors should also pay attention to the verbal time: for instance they should not use “will focus” in the abstract but rather “focused” because the review is already done.

We are grateful for your kind comments. We fixed language mistakes in the revised version of the manuscript.

Regardless, the work is scientifically sound, the schematics provided by the authors are original and very interesting. There are also somethings missing from the article formatting but the authors may have know that already since they left those sections highlighted in yellow.

Thanks for your comment. Changes have been made according to the editorial instructions.

Reviewer 3 Report

Dear Authors, I send you my comments

1) Please add the type of the review and the method use in the selection of the reference

2) Table 1 please separate experimental studies from clinical studies

3) Clinical data must be added in section 5 the authors described their experiment in animal model, so tghis section must be deleted

4) Conclusion in this review the authors described only few clinical data so it is not  possible to write that the study "suggest that the delivery of high-dose inhaled NO for its significant antimicrobial activity is safe and has promising clinical therapeutic applications in the treatment of lung infections"  

Author Response

Toronto, May 21st, 2022

To the Editor of Biomedicines

Biomedicines-1726323: “High doses of inhaled nitric oxide as an innovative antimicrobial strategy for lung infections”

Dear Editor, Dear Reviewer,

Many thanks for your thoughtful comments and review of our work.

We have revised the manuscript according to your concerns.

Please find below our point-by-point response to your review letter. We are most grateful for the possibility to revise the manuscript. We hope that all changes and corrections are in accordance with your concerns, which were very helpful in improving the quality of our work.

Sincerely yours,

Vinicius Schenk Michaelsen

REVIEWER 3

1) Please add the type of the review and the method use in the selection of the reference

We thank the reviewer for bringing this issue to our attention. This manuscript is a narrative review, and this has been specified in the text (see Introduction).

2) Table 1 please separate experimental studies from clinical studies

In Table 1 we provided the list of selected experimental studies on the antibacterial effect of nitric oxide in chronological order, to better highlight the progressive development of this potential novel therapeutic strategy. Separating clinical experimental studies from studies conducted in vitro and in vivo in animal models would not be in line with the scope we envisioned for this table. However, we removed from the table the studies that focused on testing safety of nitric oxide rather than its antibacterial activity.

3) Clinical data must be added in section 5 the authors described their experiment in animal model, so this section must be deleted

Section 5 includes the description of our studies, aiming at demonstrating for the first time the safety of the continuous administration of high-dose iNO in vivo. As stated at the end of the section, these data provide a solid base for the design of a clinical trial in patients testing the safety, feasibility and eventually efficacy of this proposed iNO administration regimen. We hence decided to keep the section unchanged.

4) Conclusion in this review the authors described only few clinical data so it is not possible to write that the study "suggest that the delivery of high-dose inhaled NO for its significant antimicrobial activity is safe and has promising clinical therapeutic applications in the treatment of lung infections"

Our narrative review described the rationale and the progressive investigational steps exploring the possibility of delivering high-dose iNO as antimicrobial treatment. As the evidence described in the review are not conclusive, we correctly stated that “the described studies suggest (do not demonstrate) that the delivery of high-dose inhaled NO….”.